# Comparison of Mortality Prediction Between the Model for End-Stage Liver Disease-3.0 (MELD-3.0) and the Model for End-Stage Liver Disease-Lactate (MELD-La) in Korean Patients with Liver Cirrhosis

**DOI:** 10.3390/medicina61030494

**Published:** 2025-03-13

**Authors:** Seung-Kang Yoo, Jeong-Han Kim, Won-Hyeok Choe, So-Young Kwon

**Affiliations:** 1Department of Internal Medicine, Konkuk University Medical Center, Seoul 05030, Republic of Korea; seungkangyoo@gmail.com (S.-K.Y.); 20050101@kuh.ac.kr (W.-H.C.); sykwonmd@kuh.ac.kr (S.-Y.K.); 2Research Institute of Medical Science, Konkuk University School of Medicine, Seoul 05029, Republic of Korea

**Keywords:** MELD, MELD-lactate, MELD-3.0, mortality prediction, liver cirrhosis

## Abstract

*Background and Objectives*: The Model for End-Stage Liver Disease (MELD) score has widely been used for mortality prediction in liver cirrhosis (LC) patients and transplantation allocation. There have been recent modifications of MELD scores, such as MELD-Lactate (MELD-La) and MELD-3.0. The goal of this study was to compare MELD, MELD-La, and MELD-3.0 in predicting mortality among LC patients in Korea. *Materials and Methods*: This is a retrospective, single-centered study in which LC patients admitted to Konkuk University Hospital between January 2011 and December 2022 were enrolled and reviewed. Predictive values for 1- and 3-month mortality for MELD, MELD-La, and MELD-3.0 were calculated using the area under the receiver operating characteristic (AUROC) curve. Differences between AUROCs were statistically analyzed using DeLong’s test. *Results*: A total of 1152 patients were initially included in this study. Among them, 165 (14.3%) patients died within one month, and 211 (19.7%) died within three months. The AUROCs for 1-month mortality of MELD, MELD-La, and MELD-3.0 were 0.808, 0.79, and 0.807, respectively. For the 3-month mortality of MELD, MELD-La, and MELD-3.0, the AUROCs were 0.805, 0.753, and 0.817, respectively. Multiple comparisons of ROC curves demonstrated that MELD and MELD-3.0 reflected the 3-month mortality prediction of LC patients better than MELD-La (*p* = 0.0018, *p* = 0.0003, respectively). *Conclusions*: This study demonstrated that MELD and MELD-3.0 outperformed MELD-La in predicting the 3-month mortality for LC patients. However, there was no significant difference between MELD and MELD-3.0 in predicting LC patient mortality.

## 1. Introduction

The Model for End-Stage Liver Disease (MELD) was initially developed to predict survival in cirrhotic patients undergoing transjugular intrahepatic portosystemic shunts (TIPS) [1]. Since then, it has been adapted for mortality prediction in liver cirrhosis (LC) patients and for determining liver transplantation priority [2,3]. The MELD score incorporates creatinine, total bilirubin, and international normalized ratio (INR) of prothrombin time, producing a score from six to 40. Higher MELD scores indicate greater mortality risk, signaling the need for urgent liver transplantation [4].

The MELD score has been continually modified to improve its accuracy by incorporating additional variables, such as sodium (MELD-Na) [5] or lactate (MELD-La) [6]. In the United States, MELD-Na has replaced the original MELD score for determining liver transplantation priority [7]. In 2020, the MELD-GRAIL-Na model was introduced, integrating re-estimated bilirubin, INR, sodium, and glomerular filtration rate assessment in liver disease (GRAIL) into a prediction of liver transplant waitlist mortality [8]. Furthermore, in 2021, MELD-3.0 was introduced, incorporating creatinine, total bilirubin, and INR with albumin, sodium, and female factors in its calculation [9]. These evolving models aim to enhance mortality prediction in LC patients.

With the emergence of many MELD score variations, numerous studies and analyses have been conducted to determine which model best predicts mortality in LC patients. Some studies advocate the superiority of MELD-La over MELD in predicting poor outcomes in LC patients with variceal bleeding [10]. Another study found that MELD-3.0 outperformed MELD-Na in mortality prediction [9]. Similarly, several studies have compared different MELD scores [11,12,13]. However, more consecutive studies are required to definitively establish which MELD variation offers the best mortality prediction.

Therefore, the goal of this study was to compare MELD, MELD-La, and MELD-3.0 in predicting mortality among LC patients in Korea.

## 2. Materials and Methods

Patients were included if their MELD, MELD-La, and MELD-3.0 scores could be calculated based on initial laboratory results from the day of admission.

A total of 1325 patients were initially enrolled. Seventy-nine patients were excluded due to a diagnosis of hepatocellular carcinoma (HCC) since HCC can increase lactate production via the Warburg effect, potentially leading to discrepancies in MELD-La calculation [14]. Patients who were lost to follow-up or underwent liver transplantation during the follow-up period were excluded.

After one month of follow-up, 1152 patients remained in the study. Over the subsequent three-month follow-up period, 78 additional patients were lost to follow-up, and three patients underwent liver transplantation. Consequently, an additional 81 patients were excluded, leaving 1071 patients for the final analysis at the three-month follow-up. A flowchart illustrating patient inclusion is presented in Figure 1.

### 2.1. Calculation of MELD, MELD-La, and MELD-3.0

The MELD, MELD-La, and MELD-3.0 scores were calculated using previously established formulas.

The MELD score was calculated using the following formula [3,15]:MELD = 9.57 × log_e_ (creatinine, mg/dL)                                           + 3.78 × log_e_ (bilirubin, mg/dL) + 11.20 × log_e_ (INR) + 6.43
where INR, creatinine, and bilirubin values lower than 1 were reported as 1. Serum creatinine values of 4 or higher were reported as 4 [3,15]:

The MELD-La score was calculated using the following formula [6]:MELD-La = 2.68 + 5.68 × log_e_ (lactate) + 0.64 × (original MELD)

For the MELD-La, all variables less than 1 for lactate are reported as 1 [6].

The MELD-3.0 score was calculated using the following formula [9]:MELD-3.0 = 1.33 (if female) + [4.56 log_e_ (bilirubin)]              + [0.82 × (137 − Na)] − [0.24 (137 − Na) × log_e_ (bilirubin)]  + [9.09 log_e_ (INR)] + [11.14 log_e_ (creatinine)]+ [1.85 × (3.5 − albumin)]                − [1.83 × (3.5 − albumin) log_e_ (creatinine)] + 6

Unlike in MELD and MELD-La, creatinine values greater than 3.0 mg/dL were reported as 3.0 mg/dL in the calculation of MELD-3.0 [9]. However, creatinine, bilirubin, and INR values lower than 1 were reported as 1, maintaining consistency with MELD and MELD-La [3,6,9,15].

### 2.2. Statistical Analysis

After calculating MELD, MELD-La, and MELD 3.0 values for the included population, the predictive accuracy for 1- and 3-month mortality was assessed using the area under the receiver operating characteristic (AUROC) curve. Statistical analyses were conducted using MedCalc software (version 22.009, MedCalc Software bvba, Ostend, Belgium). Their differences were statistically analyzed using DeLong’s test, where *p*-values less than 0.05 were considered statistically significant.

## 3. Results

### 3.1. Reasons for Hospital Visits

The reasons for hospital visits by the enrolled LC patients are shown in Figure 2. Among the 1246 patients, the most common reason for hospital visits was variceal bleeding (32.3%), followed by supportive care for general weakness, pain, and other related symptoms (25.6%) and hepatic encephalopathy (20.1%) (Figure 2).

### 3.2. Baseline Characteristics

Table 1 outlines the baseline characteristics of the study population (N = 1152). A majority of the patients were male, with a median age of 47. Alcohol was the leading cause of LC (N = 795, 69%), followed by HBV (N = 79, 6.9%).

The median total bilirubin and albumin levels were 2.55 mg/dL and 2.9 g/dL, respectively, and the median creatinine level was 0.91 mg/dL. The median values for sodium, PT INR, and lactate were 135 mEq/L, 1.43, and 2.99 mmol/L, respectively. Additionally, the median MELD, MELD-La, and MELD 3.0 were 15.74, 15.12, and 19.07, respectively.

Table 1 also presents baseline data for patients who died and those who survived after 1 month. It is noteworthy that those who died within the 1-month follow-up had higher mean lactate values (6.3 mmol/L vs. 2.82 mmol/L, *p* < 0.001) and higher respective MELD scores (24.66 vs. 14.6, *p* < 0.001) compared to those who survived.

The baseline characteristics for 3-month deaths and survivors are shown in Table 2. Similar to 1-month deaths and survivors, those who died within 3-month follow-up had higher mean lactate values (4.38 mmol/L vs. 2.81 mmol/L, *p* < 0.001). Additionally, the deceased patients exhibited higher respective MELD (23.6 vs. 13.85, *p* < 0.001), MELD-La (20.19 vs. 14.41, *p* < 0.001), and MELD-3.0 (27.57 vs. 16.84, *p* < 0.001) scores.

### 3.3. Clinical Courses of Patients

The causes of death within one and three months of the initial hospital visit are illustrated in Table 3. A total of 165 patients (14.3%) died within one month of the hospital visit. Among them, 52 (31.5%) died of hepatic failure, and 38 died of varix bleeding (23.0%), and 32 (19.4%) died from diseases other than liver disease.

A total of 211 patients (19.7%) died within three months of the hospital visit. Among them, 72 (34.1%) died due to hepatic failure, 44 (20.9%) died of varix bleeding, and 43 (20.4%) died due to diseases other than liver disease.

### 3.4. AUROC and Multiple Comparison of Different MELD Scores

#### 3.4.1. 1-Month Mortality

The receiver operating characteristic curve (ROC) and the area under curve (AUC) of MELD, MELD-La, and MELD 3.0 for 1-month mortality are shown in Figure 3a and Table 4. The AUROCs of MELD, MELD-La, and MELD 3.0 were 0.808 (*p* < 0.0001), 0.790 (*p* < 0.0001), and 0.807 (*p* < 0.0001), respectively. When the AUROCs were compared using DeLong’s test, the comparison of the three different MELD scores revealed no significant differences. For instance, the AUROCs of MELD had no significant difference compared to MELD-La (*p* = 0.8376) and MELD 3.0 (*p* = 1.000). Additionally, there were no significant differences in AUROCs between MELD-La and MELD 3.0 (*p* = 0.9633).

#### 3.4.2. 3-Month Mortality

AUROCs of MELD, MELD-La, and MELD 3.0 for 3-month mortality are also analyzed in Figure 3b and Table 3. The AUROCs of MELD, MELD-La, and MELD 3.0 were 0.805 (*p* < 0.0001), 0.753 (*p* < 0.0001), and 0.817 (*p* < 0.0001), respectively. The analyses showed that MELD outperformed MELD-La in its 3-month mortality prediction (*p* = 0.0018). Additionally, MELD 3.0 was superior to MELD-La in 3-month mortality prediction (*p* = 0.0003). However, the comparison between AUROCs of MELD and MELD 3.0 revealed no significant differences (*p* = 0.1425).

#### 3.4.3. Mortality Predictions of Different MELD Scores for Patients with Varix Bleeding

Since varix bleeding was one of the most common reasons for hospital visits, AUROCs of MELD, MELD-La, and MELD 3.0 for 1-month and 3-month mortality of patients who presented with varix bleeding are analyzed in Figure 3c,d, and Table 4. The AUROCs of MELD, MELD-La, and MELD 3.0 for 1-month mortality were 0.826 (*p* < 0.0001), 0.854 (*p* < 0.0001), and 0.823 (*p* < 0.0001), respectively. For 3-month mortality, the AUROCs were 0.831 (*p* < 0.0001), 0.841 (*p* < 0.0001), and 0.837 (*p* < 0.0001), respectively. Their comparisons, however, did not show any significant differences in both 1-month and 3-month mortality predictions for MELD, MELD-La, and MELD 3.0 (*p* = 1.0000).

## 4. Discussion

### 4.1. AUROCs for Different MELD Variations

Our study compared the predictive accuracy of MELD, MELD-La, and MELD-3.0 for 1- and 3-month mortality in Korean LC patients. In our study, the AUROCs of MELD and MELD-3.0 for 1- and 3-month mortality were greater than 0.8 (*p* < 0.0001) in total patients or variceal bleeding patients. By contrast, MELD-La demonstrated AUROCs greater than 0.8 (*p* < 0.0001) only in variceal bleeding patients but did not exceed 0.8 in the total patient cohort. Therefore, we could suggest that while MELD-La performs well in predicting outcomes for variceal bleeding patients, its overall predictive power may be limited in cirrhosis patients with varied clinical characteristics.

### 4.2. Comparison of MELD Variations in Previous Studies

The development of various MELD-based scoring systems has led to multiple studies to determine which model most effectively predicts mortality in LC patients. A previous U.S. study introduced MELD-3.0 as an improvement over the original MELD and MELD-Na, demonstrating superior predictive accuracy for mortality in LC patients [9]. Similarly, a recent study in Korea comparing four different MELD-based scores found that MELD-3.0, when combined with albumin levels, provided a more accurate prediction of 3-month survival than MELD-Na or the original MELD [16]. Furthermore, a Korean study attempted to validate the predictive performance of MELD-3.0 in alcoholic cirrhosis patients with acute decompensation. However, its superiority over the previous MELD scores was statistically insignificant [17]. Another study demonstrated that MELD-3.0 was more effective than MELD and MELD-Na in predicting 3- and 6-month mortality in Korean LC patients [18].

Numerous studies have compared the original MELD with newly developed MELD scores such as MELD-3.0, MELD-La, and MELD-GRAIL-Na [19], highlighting their potential advantages over one another. However, to our knowledge, this was the first study to directly compare MELD-3.0 and MELD-La in predicting short-term mortality among liver cirrhosis patients. Therefore, our findings provide important insights into their relative predictive performance, particularly in a Korean cohort, where region-specific validation is essential for optimizing risk stratification and transplant prioritization.

### 4.3. Rationale for Comparing MELD-La and MELD-3.0

Given that MELD-Na has already been extensively evaluated in prior studies, our study aimed to provide additional insights by comparing MELD-3.0 with MELD-La in a Korean cohort. While MELD-Na has been widely recognized as a significant modification of the original MELD [18], the clinical relevance of MELD-La remains an area of interest, particularly in a Korean cohort. By evaluating MELD-3.0 compared to MELD and MELD-La, our study provides a complementary perspective on risk stratification beyond transplant prioritization.

### 4.4. Comparison of Mortality Prediction

By comparing different MELD scores for 1-month and 3-month mortality, this study yielded some significant findings. The comparison of MELD scores for 1-month mortality did not reveal any statistically significant differences. However, for 3-month mortality, MELD-3.0 demonstrated superior predictive ability compared to MELD-La (*p* = 0.0003). Additionally, MELD showed superior predictive performance over MELD-La in 3-month mortality (*p* = 0.0018). On the other hand, MELD and MELD-3.0 showed no significant difference in predictive ability relative to each other (*p* = 0.1425).

Among patients with variceal bleeding, MELD-La and MELD showed no statistically significant difference in 1-month mortality prediction. These results contrast with a previous study that found MELD-La to have excellent discrimination for 1-month mortality in variceal bleeding patients compared to MELD [10]. These discrepancies may be attributed to differences in the study cohorts, including sample size, ethnicity, and the underlying etiology of LC. Therefore, further studies are required to better understand these differences and validate our findings.

### 4.5. Strength and Advantages

This study has several strengths. First, it focuses exclusively on an inpatient population, representing a high-risk group where mortality prediction is most clinically relevant. By limiting our analysis to inpatients, we ensured that the data are comprehensive and standardized. This approach also ensures that the findings directly apply to acute care settings where prognostic scoring systems such as MELD could be used. This approach avoids potential dilution of the results by excluding lower-risk outpatient populations, thereby providing a clearer assessment of the utility and performance of mortality prediction tools in critically ill patients.

Additionally, by including only Korean patients, we enhanced our understanding of the prognostic accuracy of MELD-based models in this population, ensuring that our findings are directly relevant to clinical decision-making in Korean LC patients.

### 4.6. Limitations and Future Research

Despite its strengths, our study has limitations. First, this was a retrospective study performed in a single center, which may limit its generalizability. Additionally, as shown in Figure 2, a considerable number of patients were admitted for supportive care (i.e., pain control, general weakness, and malnutrition management), which may have introduced heterogeneity in disease severity, potentially impacting the predictive performance of MELD-based models.

Furthermore, the relatively low proportion of female participants in our study could have influenced the results, particularly given that MELD-3.0 incorporates an adjustment for female patients. The gender imbalance could have contributed to the lack of a significant difference between MELD-3.0 and MELD as predictors of mortality in our cohort.

To address these limitations, future research should include a larger, multi-center study to further refine our understanding of the Korean LC population and improve the applicability of MELD-based scoring systems. Additionally, future studies should consider excluding patients admitted solely for supportive care, thereby refining the sample pool and ensuring more robust comparisons of mortality prediction models. Further research should also examine gender-based differences in MELD-3.0’s predictive performance using a larger, more balanced cohort to determine the impact of sex adjustments on mortality prediction.

## 5. Conclusions

In conclusion, MELD and MELD-3.0 outperform MELD-La in predicting 3-month mortality in Korean LC patients. However, MELD and MELD-3.0 showed no significant difference in their predictive ability for 3-month mortality. These findings suggest that while MELD-3.0 may offer improved predictive accuracy over MELD-La in Korean cirrhosis patients, its clinical advantage remains modest, requiring further validation.

MELD-La, on the other hand, demonstrated strong performance in patients with variceal bleeding but lacks generalizability in cirrhosis patients with varying etiologies. Future studies should evaluate whether a stratified approach—using MELD-3.0 for general risk assessment and MELD-La selectively in variceal bleeding—could enhance clinical decision-making in liver transplantation and acute care settings.

## Figures and Tables

**Figure 1 medicina-61-00494-f001:**
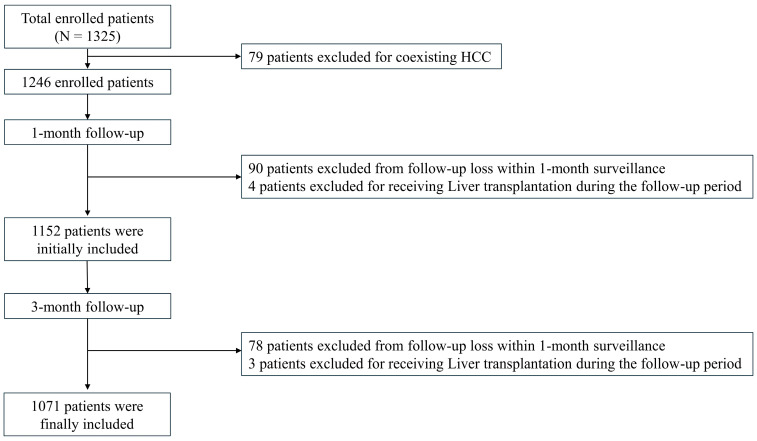
Flow chart of the sample inclusion.

**Figure 2 medicina-61-00494-f002:**
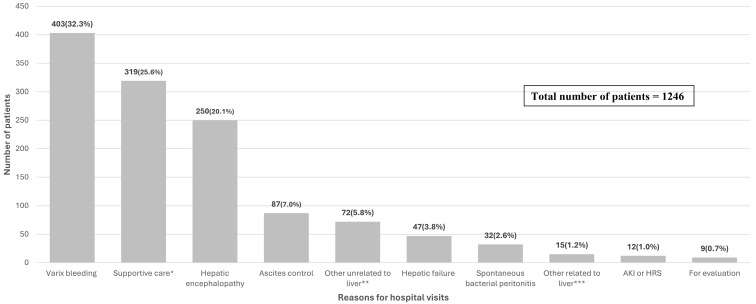
The reasons for hospital visits of enrolled patients. (N = 1246). * Supportive care related to pain control, nutritional support, and admission to general supportive care. ** Refers to diseases such as pneumonia, urinary tract infection, diverticulitis, gastric or duodenal ulcers, Mallory-Weiss tears, etc. *** Refers to jaundice, hydrothorax, alcoholic hepatitis, delirium tremens, and other diseases related to liver cirrhosis or have newly occurred in the liver.

**Figure 3 medicina-61-00494-f003:**
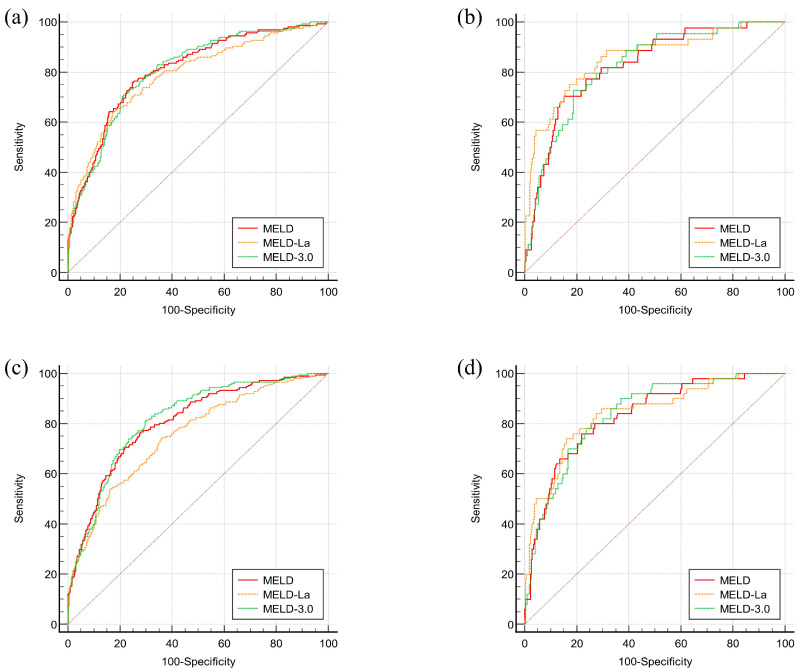
ROC curves for mortality prediction of different MELD scores in 1-month and 3-month follow-up. (**a**) ROC curve for mortality within 1 month (N = 1152), (**b**) ROC curve for mortality within 3 months (N = 1071), (**c**) ROC curve for mortality within 1 month (N = 380), (**d**) ROC curve for mortality within 3 months (N = 361).

**Table 1 medicina-61-00494-t001:** Baseline characteristics of the included patients, 1-month deaths, and 1-month survivors.

	Total Patients (N = 1152)	1-Month Deaths (N = 165)	1-Month Survivors (N = 987)	*p*-Value
Male (N, %)	790 (68.6)	125 (75.8)	665 (67.4)	*p* = 0.032
Median age (IQR)	56 (47, 65)	58 (51, 68)	56 (47, 65)	*p* = 0.02
Age (mean ± SD)	56.6 ± 13.01	58.83 ± 12.09	56.22 ± 13.13	*p* = 0.017
Etiology (N, %)				*p* = 0.963
HBV	79 (6.9)	11 (6.7)	68 (6.9)	
HCV	35 (3.0)	5 (3.0)	30 (3.0)	
Alcohol	795 (69.0)	114 (69.1)	681 (69.0)	
HBV+ Alcohol	61 (5.3)	8 (4.8)	53 (5.4)	
HCV+ Alcohol	22 (1.9)	3 (1.8)	19 (1.9)	
Autoimmune	33 (2.9)	6 (3.6)	27 (2.7)	
PBC	8 (0.7)	0 (0.0)	8 (0.8)	
NAFLD/NASH	3 (0.3)	0 (0.0)	3 (0.3)	
Unknown	116 (10.1)	18 (10.9)	98 (9.9)	
Ascites (N, %)				*p* < 0.001
Absent	460 (39.9)	26 (15.8)	434 (44.0)	
Mild to moderate	418 (36.3)	64 (38.8)	354 (35.9)	
Severe	274 (23.8)	75 (45.5)	199 (20.2)	
Encephalopathy (N, %)				*p* = 0.968
None	852 (74.0)	121 (73.3)	731 (74.1)	
Gr. 1~2	268 (23.3)	39 (23.6)	229 (23.2)	
Gr. 3~4	32 (2.8)	5 (3.0)	27 (2.7)	
Laboratory tests (IQR)				
Total bilirubin (mg/dL)	2.55 (1.37, 5.44)	5.32 (2.53, 11.8)	2.33 (1.23, 4.75)	*p* < 0.001
Albumin (g/dL)	2.9 (2.5, 3.4)	2.5 (2.2, 2.8)	3 (2.6, 3.4)	*p* < 0.001
Creatinine (mg/dL)	0.91 (0.7, 1.31)	1.42 (0.96, 2.48)	0.86 (0.68, 1.22)	*p* < 0.001
Sodium (mEq/L)	135 (130, 138)	131 (126, 136)	135 (131, 138)	*p* < 0.001
Prothrombin time (INR)	1.43 (1.25, 1.81)	2.05 (1.61, 2.68)	1.37 (1.23, 1.66)	*p* < 0.001
Lactate (mmol/L)	2.99 (2.05, 5.62)	6.3 (2.6, 13.58)	2.82 (2, 4.78)	*p* < 0.001
MELD scores (IQR)				
MELD score	15.74 (11.53, 21.72)	24.66 (19.68, 31.56)	14.6 (11.04, 19.59)	*p* < 0.001
MELD-La score	15.12 (12.49, 19.43)	22.19 (17.12, 29.69)	14.6 (12.31, 18.02)	*p* < 0.001
MELD-3.0 score	19.07 (13.02, 26.1)	28.21 (23.37, 36.08)	17.71 (12.49, 23.99)	*p* < 0.001

IQR, interquartile range; HBV, hepatitis B virus; HCV, hepatitis C virus; INR, international normalized ratio; PBC, primary biliary cirrhosis; NAFLD, Non-alcoholic fatty liver disease; NASH, Nonalcoholic Steatohepatitis; MELD, model for end-stage liver disease. Continuous values are presented by mean ± SD or median (Q1, Q3) and tested by independent *t*-test or Wilcoxon rank-sum test. Categorical values are presented by N (%) and tested by a chi-squared test.

**Table 2 medicina-61-00494-t002:** Baseline characteristics of the 3-month deaths and survivors.

	Total Patients (N = 1071)	3-Month Deaths (N = 211)	3-Month Survivors (N = 860)	*p*-Value
Male (N, %)	732 (68.3)	155 (73.5)	577 (67.1)	*p* = 0.075
Median age (IQR)	56 (47.5, 65)	59 (51, 67.5)	56 (47, 64)	*p* = 0.002
Age (mean ± SD)	56.59 ± 13.05	59.02 ± 11.92	56 ± 13.25	*p* = 0.003
Etiology (N, %)				*p* = 0.232
HBV	72 (6.7)	12 (5.7)	60 (7.0)	
HCV	33 (3.1)	6 (2.8)	27 (3.1)	
Alcohol	737 (68.8)	143 (67.8)	594 (69.1)	
HBV+ Alcohol	56 (5.2)	9 (4.3)	47 (5.5)	
HCV+ Alcohol	22 (2.1)	9 (4.3)	13 (1.5)	
Autoimmune	30 (2.8)	7 (3.3)	23 (2.7)	
PBC	8 (0.7)	0 (0.0)	8 (0.9)	
NAFLD/NASH	2 (0.2)	0 (0.0)	2 (0.2)	
Unknown	72 (6.7)	12 (5.7)	60 (7.0)	
Ascites (N, %)				*p* < 0.001
Absent	429 (40.1)	30 (14.2)	399 (46.4)	
Mild to moderate	387 (36.1)	85 (40.3)	302 (35.1)	
Severe	255 (23.8)	96 (45.5)	159 (18.5)	
Encephalopathy (N, %)				*p* = 0.901
None	803 (75.0)	158 (74.9)	645 (75.0)	
Gr. 1~2	238 (22.2)	48 (22.7)	190 (22.1)	
Gr. 3~4	30 (2.8)	5 (2.4)	25 (2.9)	
Laboratory tests (IQR)				
Total bilirubin (mg/dL)	2.5 (1.33, 5.26)	5.14 (2.56, 10.48)	2.15 (1.2, 4.27)	*p* < 0.001
Albumin (g/dL)	2.9 (2.5, 3.4)	2.6 (2.2, 2.9)	3 (2.6, 3.5)	*p* < 0.001
Creatinine (mg/dL)	0.91 (0.7, 1.31)	1.39 (0.96, 2.12)	0.85 (0.68, 1.13)	*p* < 0.001
Sodium (mEq/L)	135 (130, 138)	131 (126, 135)	136 (131, 138.25)	*p* < 0.001
Prothrombin time (INR)	1.42 (1.24, 1.78)	1.94 (1.48, 2.49)	1.35 (1.22, 1.6)	*p* < 0.001
Lactate (mmol/L)	2.97 (2.03, 5.79)	4.38 (2.44, 10.94)	2.81 (2, 4.9)	*p* < 0.001
MELD scores (IQR)				
MELD score	15.46 (11.37, 21.25)	23.6 (18.07, 30.09)	13.85 (10.73, 18.76)	*p* < 0.001
MELD-La score	15.04 (12.43, 19.3)	20.19 (15.71, 27.49)	14.42 (12.14, 17.88)	*p* < 0.001
MELD-3.0 score	18.84 (12.86, 25.7)	27.57 (22.87, 33.63)	16.84 (12.09, 22.8)	*p* < 0.001

IQR, interquartile range; HBV, hepatitis B virus; HCV, hepatitis C virus; INR, international normalized ratio; PBC, primary biliary cirrhosis; NAFLD, Non-alcoholic fatty liver disease; NASH, Nonalcoholic Steatohepatitis; MELD, model for end-stage liver disease. Continuous values are presented by mean ± SD or median (Q1, Q3) and tested by independent *t*-test or Wilcoxon rank-sum test. Categorical values are presented by N (%) and tested by a chi-squared test.

**Table 3 medicina-61-00494-t003:** Causes of death of those who died within the 1- and 3-month follow-up.

Variables	1-Month Deaths (N, %)	3-Month Deaths (N, %)
Cause of Death		
Varix bleeding	38 (23.0%)	44 (20.9%)
Hepatic encephalopathy	7 (4.2%)	11 (5.2%)
Spontaneous bacterial peritonitis	11 (6.7%)	14 (6.6%)
Hepatorenal syndrome	15 (9.1%)	15 (7.1%)
Hepatic failure	52 (31.5%)	72(34.1%)
Other than liver disease	32 (19.4%)	43 (20.4%)
Unknown	10 (6.1%)	12 (5.7%)
Total deaths	165 (100%)	211 (100%)

**Table 4 medicina-61-00494-t004:** Comparison of AUROCs for 1- and 3-month mortality in sample populations.

Category	Time Period	MELD	MELD-La	MELD-3.0
Total patients	1 month (N = 1152)	0.808 (*p* < 0.0001)	0.790 (*p* < 0.0001)	0.807 (*p* < 0.0001)
		Pairwise comparison of AUROCs - MELD vs. MELD-La: *p* = 0.8376 - MELD vs. MELD-3.0: *p* = 1.0000 - MELD-La vs. MELD-3.0: *p* = 0.9633		
	3 months (N = 1071)	0.805 (*p* < 0.0001)	0.753 (*p* < 0.0001)	0.817 (*p* < 0.0001)
		Pairwise comparison of AUROCs - MELD vs. MELD-La: *p* = 0.0018 - MELD vs. MELD-3.0: *p* = 0.1407 - MELD-La vs. MELD-3.0: *p* = 0.0003		
Varix bleeding	1 month (N = 380)	0.826 (*p* < 0.0001)	0.854 (*p* < 0.0001)	0.823 (*p* < 0.0001)
		Pairwise comparison of AUROCs - MELD vs. MELD-La: *p* = 1.0000 - MELD vs. MELD-3.0: *p* = 1.0000 - MELD-La vs. MELD-3.0: *p* = 1.0000		
	3 months (N = 361)	0.831 (*p* < 0.0001)	0.841 (*p* < 0.0001)	0.837 (*p* < 0.0001)
		Pairwise comparison of AUROCs - MELD vs. MELD-La: *p* = 1.0000 - MELD vs. MELD-3.0: *p* = 1.0000 - MELD-La vs. MELD-3.0: *p* = 1.0000		

## Data Availability

Data is contained within the article.

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
