# Peer review of "Comparison of Mortality Prediction Between the Model for End-Stage Liver Disease-3.0 (MELD-3.0) and the Model for End-Stage Liver Disease-Lactate (MELD-La) in Korean Patients with Liver Cirrhosis"

_medicina, 2025, doi:10.3390/medicina61030494_

Round 1

Reviewer 1 Report

Comments and Suggestions for Authors

This is a retrospective study comparing MELD, MELD 3.0 and MELD-Lactate to predict 1 and 3 month survival in 1152 hospitalized Korean patients with liver cirrhosis of variable etiologies.
The authors conclude that MELD 3.0 and MELD are equally effective at predicting 3 month mortality in these patients and significantly better than MELD-Lactate. The original MELD score is no longer used to allocate organs in the US, it was replaced by the MELD-Na and then more recently replaced by the MELD 3.0 (as of June 2024). . When performing a quick review of the literature however it does appear that there is a larger retrospective study from Korea comparing the MELD 3.0 with  MELD-Na which is a more updated version of the MELD score and was until recently the primary tool for organ allocation. That study found that the MELD 3.0 outperformed the MELD-Na as a predictor of 3 month mortality which is in line with previously published data in US patients. I am not sure that comparing the original MELD to the MELD. 3.0 adds much to existing knowledge. Interestingly this study did not find a significant difference between the MELD 3.0 and MELD as predictors of mortality which may be due to the relatively small number of female patients in the study (the MELD 3.0 particularly advantages  female patients). Furthermore, the MELD Lactate score is mostly relevant in predicting morality in critically ill patients, particularly those with variceal bleeding and is otherwise not widely used. As the authors themselves note, there was a substantial number of patients hospitalized for supportive measures which may have contributed to the lower AUROC for the MELD-Lactate. 
Overall, I think the authors need to better explain why their findings are relevant and how they add to existing literature. The original MELD is no longer used (at least not in the US) and there is already data from a much larger sample of Korean patients showing superiority of the MELD 3.0 vs MELD-Na, a more updated version of the MELD.  In its current form I do not think this paper adds much to existing knowledge. 

Comments on the Quality of English Language

needs some minor grammatical edits

Author Response

Comment #1 ) The authors conclude that MELD 3.0 and MELD are equally effective at predicting 3 month mortality in these patients and significantly better than MELD-Lactate. The original MELD score is no longer used to allocate organs in the US, it was replaced by the MELD-Na and then more recently replaced by the MELD 3.0 (as of June 2024).

Response #1) Thank you for your comment. While it is true that MELD-Na replaced the original MELD for transplantation allocation in the United States, the original MELD is still widely being used in clinical practice for non-transplant patients, particularly in acute care settings. Many centers, including those in Korea, still use MELD as a risk stratification tool for cirrhotic patients, regardless of its role in transplantation.

Comment #2) When performing a quick review of the literature however it does appear that there is a larger retrospective study from Korea comparing the MELD 3.0 with MELD-Na which is a more updated version of the MELD score and was until recently the primary tool for organ allocation. That study found that the MELD 3.0 outperformed the MELD-Na as a predictor of 3 month mortality which is in line with previously published data in US patients. I am not sure that comparing the original MELD to the MELD. 3.0 adds much to existing knowledge.

Response #2) We appreciate your thoughtful observation. We acknowledge that MELD-Na has been widely studied as a modification of the original MELD score. Additionally, we recognize that prior studies, including a large retrospective study in Korea, have demonstrated the superiority of MELD-3.0 over MELD-Na. However, given that MELD-Na has already been extensively evaluated, our study aimed to provide additional insights by comparing MELD-3.0 with MELD-La in Korean cohort. To clarify this, we have revised the Discussion, page 9, paragraph 4, Lines 228-235 (in green font) to explain why we chose to compare MELD-3.0 with MELD-La instead of MELD-Na.

Comment #3) Interestingly this study did not find a significant difference between the MELD 3.0 and MELD as predictors of mortality which may be due to the relatively small number of female patients in the study (the MELD 3.0 particularly advantages female patients).

Response #3) Thank you for highlighting this point. We agree that MELD-3.0 incorporates an adjustment for female patients, and the relatively low proportion of female participants in our study could have influenced the results. This could have contributed to the lack of significant difference between MELD-3.0 and MELD as predictors of mortality in our cohort. To clarify this, we have revised the Discussion section (Page 10, paragraph 6-7, Line 270-273 and 278-280, in green font) to acknowledge this as a limitation of our study.

Comment #4) Furthermore, the MELD Lactate score is mostly relevant in predicting mortality in critically ill patients, particularly those with variceal bleeding and is otherwise not widely used. As the authors themselves note, there was a substantial number of patients hospitalized for supportive measures which may have contributed to the lower AUROC for the MELD-Lactate. 

Response #4) We agree that MELD-La is mostly relevant in predicting mortality in critically ill patients, and that many patients in our cohort were hospitalized for supportive care. To address this concern, we performed a subgroup analysis comparing MELD, MELD-La, and MELD-3.0n specifically in variceal bleeding patients. We believe this analysis provides further insights into whether MELD-La retains predictive value in this subgroup, thereby mitigating the relevance concern mentioned in your comment.

Considering these points, we strongly believe that while MELD-Na has been extensively studied, our comparison of MELD, MELD-La, and MELD-3.0 in Korean cohort provides a complementary insight to the existing literature.

Thank you again for your thorough review.

Reviewer 2 Report

Comments and Suggestions for Authors

Manuscript ID: medicina-3508794
Comparison of Mortality Prediction between the Model for End-Stage
Liver Disease-3.0 (MELD-3.0) and the Model for End-Stage Liver
Disease-Lactate (MELD-La) with Liver Cirrhosis in Korean Patients
by Seung-Kang Yoo, Jeong-Han Kim *, Won Hyeok Choe, So Young Kwon
Abstract:
Line 20 change 3month as 3-month in agreement as 1-month
Line 29 letter p is in italics  as p=0.0018, p=0.0003; also change p=0.0003, respectively
Materials and Methods
Line 67 change .....enrolled. 79 as …..enrolled, 79
Figure 1 need major resolution 
Statistical analysis
Line 110 letter p is in italics.
Results
Figure 2 and Figure 3 need major resolution
Line 119  change N in lowercase see N=1246 as n=1,246 in agreement with line 127; also, n is in italics
Please check all section result the use of p as italics lines 134, 138, 140,166, 167, 170, 171,176,178,179,180,186,187,189,
Lines 193 and 194  N is in lowercase and italics
In tables 1 to 3 n is in italics and P-value is italic and lowercase
Tables 1 and 2 the values of significance need the letter p in agreement with Table 3; also in table 3 N is in lowercase and italics
Discussion
Lines 203, 205,233, 236 the letter p is in italics
References
All the references they are incomplete  change et al for authors in agreement with the format of the journal see instructions for authors
Also check all reference  the abbreviated name of journals are  in italics; volume is in italics
References 8,9, 10, 11, 13 the title must be in agreement with format of  the journal see  reference 1 to 6 of manuscript. 

Author Response

Summary : Thank you very much for your time to read our manuscript. Please find the detailed response below and the corresponding revisions in the re-submitted file. The corrections were made using red font in the revised manuscript.

Comment 1: Line 20 change 3month as 3-month in agreement as 1-month.

Response: Thank you for your careful review. We have corrected "3month" to "3-month" in line 20 to maintain consistency with "1-month". This change was made using red font in the revised manuscript.

Comment 2: Line 29 letter p is in italics as p=0.0018, p=0.0003; also change p=0.0003, respectively.

Response: We have corrected the formatting of p values to italics and revised the phrase to ensure clarity. The corrections were made in red font in the revised manuscript.

Comment 3: Materials and Methods Line 67 change .....enrolled. 79 as …..enrolled, 79.

Response: This grammatical issue has been corrected as per your suggestion, and the corrections were made in red font in the revised manuscript.

Comment 4: Figure 1 need major resolution.

Response: We have replaced Figure 1 with a higher-resolution version to improve clarity. The updated figure is included in the revised manuscript.

Comment 5: Statistical analysis Line 110 letter p is in italics.

Response: We have corrected the formatting of p in line 110 to italics. The changes were made in red font in the revised manuscript.

Comment 6: Results Figure 2 and Figure 3 need major resolution.

Response: We have replaced Figures 2 and 3 with higher-resolution versions to enhance clarity. The updated figures are included in the revised manuscript.

Comment 7: Line 119 change N in lowercase see N=1246 as n=1,246 in agreement with line 127; also, n is in italics.

Response: Thank you for your comment. To maintain consistency throughout the manuscript, we have standardized the use of "N" instead of "n" across all sections, including text, tables, and figures. This ensures clarity and uniformity in reporting sample sizes. All instances of "n" have been replaced with "N" accordingly, and these changes were made in red font in the revised manuscript.

Comment 8: Please check all section result the use of p as italics lines 134, 138, 140,166, 167, 170, 171,176,178,179,180,186,187,189,.

Response: We have carefully reviewed all mentioned lines and ensured that p values are consistently italicized. These changes were made in red font in the revised manuscript.

Comment 9: Lines 193 and 194 N is in lowercase and italics.

Response: Thank you for your comment. Like we mentioned from the response of the comment 7, in order to maintain consistency throughout the manuscript, all instances of "n" have been replaced with "N" accordingly, and changes were made in red font in the revised manuscript.

Comment 10: In tables 1 to 3 n is in italics and P-value is italic and lowercase

Response: We have reviewed Tables 1 to 3 and made sure that n is italicized and p-values are italicized and lowercase throughout. These changes are reflected in red font in the revised manuscript.

Comment 11: Tables 1 and 2 the values of significance need the letter p in agreement with Table 3; also in table 3 N is in lowercase and italics.

Response: We have checked the formatting and ensured that p-values follow the same format across Tables 1, 2, and 3. Regarding N, (please refer to the responses of the comment 7, and 9) we have decided not to replace N to n in order to maintain consistency throughout the manuscript.

Comment 12: Discussion Lines 203, 205, 233, 236 the letter p is in italics

Response: We have formatted all p values in these lines to be italicized as per the journal’s style. The corrections were made in red font in the revised manuscript.

Comment 13: References All the references they are incomplete change et al for authors in agreement with the format of the journal see instructions for authors. Also check all reference the abbreviated name of journals are in italics; volume is in italics. References 8, 9, 10, 11, 13 the title must be in agreement with format of the journal see reference 1 to 6 of manuscript.

Response: Thank you very much for pointing it out. We have reviewed and revised all references to ensure they are complete. "et al." has been replaced with full author names in accordance with the journal’s formatting requirements. We have reviewed the MDPI reference guide v9 and made changes in accordance with the guideline provided. All the changes were made in red font in the revised manuscript. Also, the titles of references have been consistently formatted with capital letters at the beginning of each word, except for prepositions and “the”. This ensures that all reference titles follow the same style throughout the manuscript.

Again, Thank you very much for your thorough review of our work.

Reviewer 3 Report

Comments and Suggestions for Authors

The manuscript is not original, but provides interesting information. It is concluded that MELD and MELD 3.0 have no significant difference in predicting mortality at 1 and 3 months and that MELD LA is better at predicting mortality due to bleeding from esophageal varices.

Consequently, the results are important, because they support that the original score is as good as the MELD 3.0 and the MELD LA. The introduction, methodology, presentation of results and discussion seem appropriate to me.

I only suggest checking your English by an expert.

Comments on the Quality of English Language

I  suggest checking your English by an expert.

Author Response

Comment:

The manuscript is not original but provides interesting information. It is concluded that MELD and MELD 3.0 have no significant difference in predicting mortality at 1 and 3 months and that MELD-LA is better at predicting mortality due to bleeding from esophageal varices.

Consequently, the results are important because they support that the original score is as good as MELD 3.0 and MELD-LA. The introduction, methodology, presentation of results, and discussion seem appropriate to me.

I only suggest checking your English by an expert.

Response:
Thank you for your valuable feedback and for recognizing the significance of our study. We appreciate your suggestion regarding English language refinement. In response, we have carefully reviewed the manuscript and made grammatical and stylistic improvements to enhance clarity and readability.

Minor grammatical and stylistic adjustments were made throughout the manuscript. These revisions focus on improving sentence structure, word choice, and overall readability.

We believe that these refinements further enhance the clarity and presentation of our findings. All language-related revisions are written in blue font in the revised manuscript for easy identification.

Thank you very much for your review.